# Cannabidiol Content and In Vitro Biological Activities of Commercial Cannabidiol Oils and Hemp Seed Oils

**DOI:** 10.3390/medicines7090057

**Published:** 2020-09-07

**Authors:** Masashi Kitamura, Yuka Kiba, Ryuichiro Suzuki, Natsumi Tomida, Akemi Uwaya, Fumiyuki Isami, Shixin Deng

**Affiliations:** 1Laboratory of Pharmacognocy, School of Pharmacy, Faculty of Pharmacy and Pharmaceutical Sciences, Josai University, 1–1, Keyakidai, Sakado, Saitama 350-0295, Japan; yy17087@josai.ac.jp (Y.K.); ryu_suzu@josai.ac.jp (R.S.); 2Research and Development, Morinda Worldwide, Inc., Morinda Bldg., 3-2-2 Nishishinjuku, Shinjuku-ku, Tokyo 160-0023, Japan; Natsumi_Tomida@jp.morinda.com (N.T.); akemi_uwaya@jp.morinda.com (A.U.); fumiyuki_isami@jp.morinda.com (F.I.); 3Research and Development, Morinda Inc., 737 East 1180 South, American Fork, UT 84003, USA; Shixin_Deng@newage.com

**Keywords:** Cannabidiol, hemp, antioxidant, *Cannabis sativa*

## Abstract

**Background:** Hemp (*Cannabis sativa* L.) seed contains high contents of various nutrients, including fatty acids and proteins. Cannabidiol (CBD) is a non-psychoactive compound that can be extracted from *C. sativa* and used for treating epilepsy and pain. Industrial hemp products, including CBD and hemp seed oils, have become increasingly popular. Some products are marketed without a clear distinction between CBD and hemp seed oils. Herein, the CBD content and biological activities of commercial CBD and hemp seed oils were examined. **Methods:** CBD content was measured by high-performance liquid chromatography. For in vitro antioxidant activity determination, 2,2-diphenyl-1-picrylhydrazyl and 2,2′-azinobis (3-ethylbenzothiazoline-6-sulfonic acid) radical-scavenging assays were performed. **Results:** The CBD concentrations in the two CBD oil samples were 18.9 ± 0.5 and 9.2 ± 0.4 mg/mL. Of the seven hemp seed oil samples, six samples contained CBD in concentrations ranging from 2.0 ± 0.1 to 20.5 ± 0.5 µg/mL, but it was not detected in one sample. Antioxidant activity was observed in both CBD oil samples. **Conclusions:** The results indicate that (1) CBD content varied by hemp seed oil sample and that (2) antioxidant activity could be a useful landmark for discriminating CBD oils from hemp seed oils.

## 1. Introduction

*Cannabis sativa* L. has been cultivated worldwide for centuries for medicinal, industrial and recreational use. The characteristic compounds of *C. sativa* L. are cannabinoids, including delta-9 tetrahydrocannabinol (THC) and cannabidiol (CBD) [1]. THC is the main psychoactive compound, whereas CBD is non-psychoactive and can effectively treat epilepsy and other neuropsychiatric disorders [2,3,4,5,6,7,8]. Therefore, an increasing number of CBD products have been marketed worldwide. Commercial CBD products are primarily produced from industrial *C. sativa* or hemp, which contains little or no THC [9,10]. Hemp seed is often utilized as a functional food and medicine. In Kampo medicine, it is known as Mashinin, and is used as a laxative [11]. Hemp seed oil contains >80% polyunsaturated fatty acids with two essential fatty acids, i.e., linoleic acid (18:2 omega-6) and alpha-linolenic acid (18:3 omega-3) [12]. The omega-6 to omega-3 ratio in hemp seed oil is considered to be nutritionally optimal [9,10,12]. Some reports have demonstrated that cannabinoids can be detected in commercial hemp seed products [12,13]. Considering that hemp seed produces no cannabinoids in its metabolic pathways, it could be contaminated with resin from the flowers and leaves, which contain cannabinoids, during hemp seed processing [12,13,14].

Few reports have been published on the CBD quantification of CBD oil products marketed in Japan, or on the existence of CBD in hemp seed oils commercially available in Japan. In addition, some products are marketed without a clear distinction between CBD and hemp seed oils. Essential oil obtained from hemp is commonly studied using a multidisciplinary approach for evaluating various biological activities, including antioxidant capacity, enzyme inhibition, antimicrobial activity and cytotoxicity on cell lines [15,16,17]. CBD has demonstrated strong antioxidant effects and can be considered a potent neuroprotective agent [8,18]. Therefore, in vitro biological assays might be useful for discriminating CBD oils from hemp seed oils. Herein, commercially available CBD and hemp seed oil products were analyzed using high-performance liquid chromatography (HPLC). Additionally, their antioxidant activities were determined.

## 2. Materials and Methods

### 2.1. Samples

CBD, hemp seed and rapeseed oils were purchased online as described in Table 1.

### 2.2. HPLC Analysis

HPLC analyses were performed using an Alliance Waters e2695 (Waters) and Alliance Waters 2998 PDA detector. An XTerra MS C18 column (3.5 µm, 4.6 mm × 150 mm, Waters) was used with an isocratic mobile phase of water (0.1% formic acid): acetonitrile 3:7 at a flow rate of 0.8 mL/min for 30 min. The column temperature was set to 35 °C and sample injection volume was 10 µL. The chromatograms were monitored at 235 nm. The HPLC analysis for CBD detection and validation was based on previously reported methods with modification [12,19]. CBD and THC reference standards were obtained from Cayman Chemical, Michigan, USA and Sigma-Aldrich, USA, respectively. Standard stock solutions of CBD (100 µg/mL) were prepared in methanol, and calibration curves were prepared with the CBD standard at concentrations of 100, 50, 25, 12.5, 5 and 0.5 µg/mL (Appendix A). Linearity was assessed by evaluating the coefficient of determination (R^2^). HPLC chromatographs of CBD and THC standards are shown in Appendix A. The instrumental limit of detection (LOD) values were calculated using the following formula: LOD = (3.3 × σ)/m, where σ is the residual standard deviation of the calibration curve and m is the calibration curve slope. The limit of quantification (LOQ) values were determined according to the following formula: LOQ = (10 × σ)/m. The intra-day repeatability of the method was assessed by analyzing three replicates of the samples injected thrice on the same day. The relative standard deviation (RSD) of the three replicates was also calculated.

### 2.3. Sample Preparation for HPLC

CBD oils were diluted with methanol and injected into the HPLC system. First, 5 mL of hemp seed oil was extracted with 20 mL methanol. After solvent evaporation, the samples were heated at 100 °C for 2 h. Subsequently, the samples were added to methanol and injected into the HPLC system. All the analyses were performed in triplicate, and the data are expressed as mean ± standard deviation.

### 2.4. Antioxidant Assay

2,2-Diphenyl-1-picrylhydrazyl (DPPH) and 2,2′-azinobis (3-ethylbenzothiazoline-6-sulfonic acid) (ABTS) radical-scavenging assays were performed. The oils (250 µL) were extracted with 1 mL methanol for 10 min with occasional shaking. After incubation, the supernatant was used as the sample solution. The DPPH assay was performed according to a previously described protocol [20,21]. In a total volume of 200 µL, 180 µL of 100 µM DPPH dissolved in methanol and 20 µL of the samples dissolved in methanol were incubated for 30 min at room temperature. The absorbance of the mixture was measured at 517 nm using a microplate reader (Molecular Devices SPECTRA MAX M2). The ABTS radical-scavenging activity was determined according to a previously described assay with some minor modifications [21]. Briefly, 7 mM ABTS in H_2_O and 2.45 mM potassium persulfate were mixed in a 1:1 ratio. Then, the solution was diluted in methanol at a 1:25 ratio. In a 96-well plate, 180 µL of the diluted solution and 20 µL of each sample dissolved in methanol were mixed and incubated for 10 min at room temperature, and absorbance was measured at 734 nm. The percentage enzyme activity inhibition was calculated by comparison with the negative control: Inhibition (%) = 100 × (A_0_ − A_1_)/A_0_, where A_0_ is the negative control absorbance and A_1_ is the sample absorbance. For each sample with a percentage inhibition of >60%, the IC_50_ values were determined from a dose–response curve with eight different concentrations. Ascorbic acid was used as a positive control.

## 3. Results

### 3.1. HPLC Analysis of CBD and Hemp Seed Oils

Herein, two CBD and seven hemp seed oil samples were evaluated, while rapeseed oil was used as a CBD-free sample. To quantify the CBD concentration of CBD and hemp seed oils, HPLC analysis was performed. A calibration curve was constructed using CBD standards (100, 50, 25, 12.5, 5 and 0.5 µg/mL). The linearity plot and chromatograph are shown in the Appendix A, and the linearity coefficient of determination (R^2^) was 0.995. From Table 2, the LOD, LOQ, as well as intra- and inter-day RSD were 1.5 µg/mL, 4.6 µg/mL, 4.2% and 2.4%, respectively.

Using the HPLC method, CBD and hemp seed oil samples were analyzed. Because CBD exists primarily in the carboxylated form, cannabidiol acid (CBDA), in fresh *C. sativa* L. plants, hemp seed oil samples were heated at 100 °C for 2 h to decarboxylate CBDA into CBD [12]. The CBD concentration of each sample is shown in Figure 1. The CBD oil samples (C1 and C2) showed concentrations of 18.9 ± 0.5 and 9.2 ± 0.4 mg/mL, respectively, which were nearly equal to the value displayed on the product description (500 mg/30 mL and 300 mg/30 mL, respectively). Of the seven hemp oil samples tested, six contained CBD concentrations ranging from 2.0 ± 0.1 (C6) to 20.5 ± 0.5 (C4) µg/mL, and CBD was not detected in one hemp seed oil sample (C7) or rape oil sample (C9).

### 3.2. Biological Activities of CBD and Hemp Seed Oils

To assess the biological activities of CBD and hemp seed oils in vitro, ABTS and DPPH assays were performed (Table 3). The DPPH and ABTS assay results indicated that antioxidant activities in the CBD oil samples were higher than those in the hemp seed oil samples. The CBD oil samples (C1 and C2) showed dose–response DPPH scavenging activities with IC_50_ values of 17.9 and 18.6 µL/mL, respectively. The CBD oil samples (C1 and C2) also showed dose–response ABTS scavenging activities with IC_50_ values of 2.3 and 3.6 µL/mL, respectively. To confirm the antioxidant activity of the CBD oil samples, the DPPH and ABTS scavenging activities were evaluated using a CBD standard. The IC_50_ values of the CBD standard in the DPPH and ABTS assays were 584 and 92.2 µM, respectively. The IC_50_ values of the CBD oil samples (C1 and C2) calculated using each concentration obtained from Figure 1 were respectively 1075 and 522 µM in the DPPH assay, and 138 and 67 µM in the ABTS assay. These results indicated that the antioxidant potential of CBD oil could be attributed to CBD.

## 4. Discussion

Herein, it was found that six of seven hemp seed oil products contained CBD, while the concentration varied by sample. Therefore, the unintended intake of CBD from hemp seed oil products might occur. Considering the adverse effects of CBD, including developmental toxicity, embryo-fatal mortality and neurotoxicity, unintended consumption of CBD should be avoided [22]. The cannabinoid content in hemp seed oil products could be affected by strain, product origin and processing steps. In this study, THC was not detected in any samples. Previous reports showed that some hemp seed oil products contained cannabinoids, including CBD and THC [10,12]. In Japan, CBD and hemp seed oil products containing THC are strictly prohibited by law. In addition, some products are marketed without a clear distinction between CBD and hemp seed oils. Therefore, the quality control of CBD and hemp seed oil products is necessary. In the antioxidant assays, the CBD oils exhibited free radical scavenging activities in a CBD dose-dependent manner. Because the DPPH and ABTS assays are simple and rapid, antioxidant activity represents a useful marker for distinguishing between CBD and hemp seed oils. Previously, the antioxidant activities of cannabinoids and *C. sativa* extracts were evaluated by DPPH and ABTS assays [18]. CBD and THC showed similar antioxidant activities by ABTS assay, whereas the antioxidant activity of CBD determined by DPPH assay was two-fold higher than that of THC [18]. Therefore, the ratio of antioxidant activity determined by DPPH assay to that by ABTS assay might be useful information for the quality control of CBD oils to avoid contamination with THC.

## 5. Conclusions

In this study, we analyzed the CBD contents and antioxidant activities of CBD and hemp seed oils commercially available in Japan. The CBD contents of hemp seed oils varied by sample. Because CBD oil samples exhibited strong antioxidant activities, antioxidant assays represent a useful marker for distinguishing between CBD and hemp seed oils. Our results could be applicable for the quality control of CBD and hemp seed oil products.

## Figures and Tables

**Figure 1 medicines-07-00057-f001:**
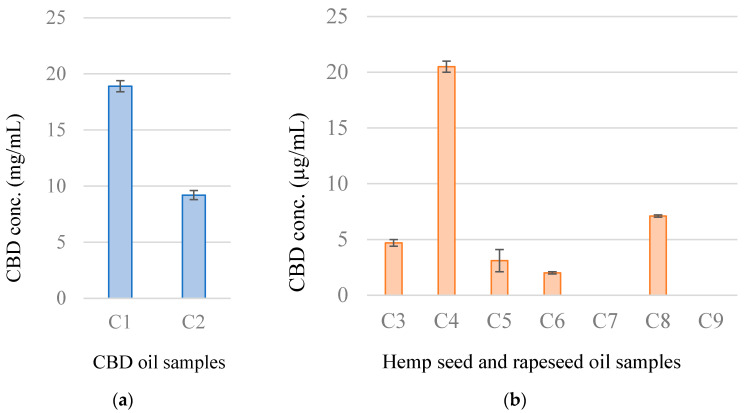
CBD concentrations in CBD oil (**a**) and hemp seed oil (**b**) samples.

**Table 1 medicines-07-00057-t001:** CBD and hemp seed oil samples in this study.

Sample	Product	Product Origin
C1	CBD oil	USA
C2	CBD oil	Unknown
C3	Hemp seed oil	Australia
C4	Hemp seed oil	New Zealand
C5	Hemp seed oil	Unknown
C6	Hemp seed oil	Germany
C7	Hemp seed oil	Germany
C8	Hemp seed oil	Canada
C9	Rapeseed oil	Japan

**Table 2 medicines-07-00057-t002:** Parameters for method validation.

Factors	Results
Equation	y = 18600x − 23348
R^2^	0.995
LOD (µg/mL)	1.5
LOQ (µg/mL)	4.6
RSD (Intra-Day)	4.2%
RSD (Inter-Day)	2.4%

**Table 3 medicines-07-00057-t003:** Biological activities of CBD and hemp seed oils.

Sample	Antioxidant Activity			
	DPPH		ABTS	
	Sample Conc. (25 µL/mL)	IC_50_ (CBD)	Sample Conc. (25 µL/mL)	IC_50_ (CBD)
C1	64.8 ± 0.8%	17.9 µL/mL (1075 µM)	90.5 ± 0.1%	2.3 µL/mL (138 µM)
C2	63.2 ± 1.4%	18.6 µL/mL (522 µM)	90.3 ± 0.1%	3.6 µL/mL (67 µM)
C3	12.7 ± 0.8%	-	28.2 ± 2.4%	-
C4	23.2 ± 1.0%	-	47.8 ± 2.0%	-
C5	19.8 ± 0.4%	-	21.8 ± 1.3%	-
C6	11.1 ± 0.6%	-	27.3 ± 1.1%	-
C7	1.3 ± 0.8%	-	2.8 ± 1.2%	-
C8	11.6 ± 1.1%	-	26.9 ± 1.0%	-
C9	14.5 ± 0.6%	-	36.6 ± 1.9%	-
Ascorbic Acid	-	38.0 µM	-	11.3 µM
CBD	-	584 µM	-	92.2 µM

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
