# Peer review of "Cannabidiol Content and In Vitro Biological Activities of Commercial Cannabidiol Oils and Hemp Seed Oils"

_medicines, 2020, doi:10.3390/medicines7090057_

Round 1
Reviewer 1 Report
I only suggest to move paragraph 3.1 and Table 2 in the Methods section.
Data about CBD concentration need to be presented in the Results section. Consider to prepare another Table with such concentrations (so, move them from Table 1).
Author Response
I would like to ask you to consider our revised manuscript entitled "Cannabidiol content and in vitro biological activities of commercial cannabidiol oils and hemp seed oils"
According to the reviewers’ comments, we revised the contents as the accompanying sheet. I would appreciate your correspondence.
According to your comment, CBD concentrations are shown in Figure1 in paragraph 3.1.
Page.4 Figure. 1
[Others]
According to the reviewers’ comments, we clarified our purpose, results and discussion.
All modifications are listed below.
- Page 1, Lane 16-17
- Some products are marketed without a clear distinction between CBD and hemp seed oils.
- Page 1, Lane 25-26
- The results indicate that (1) CBD content varied by hemp seed oil sample and that (2) antioxidant activity could be a useful landmark for discriminating CBD oils from hemp seed oils.
- Page 2, Lane 44-46
- Few reports have been published on CBD quantification of CBD oil products marketed in Japan or on the existence of CBD in hemp seed oils commercially available in Japan. In addition, some products are marketed without a clear distinction between CBD and hemp seed oils.
- Page 2, Lane 50-53
- CBD has demonstrated strong antioxidant effects and can be considered a potent neuroprotective agent [8, 18]. Therefore, in vitro biological assay might be useful for discriminating CBD oils from hemp seed oils. Herein, commercially available CBD and hemp seed oil products were analyzed using high-performance liquid chromatography (HPLC). Additionally, their antioxidant activities were determined.
- Page 2, Lane 57
- Table 1 *Deletion of CBD contents → Figure 1
- Page 2, Lane 64-65
- The HPLC analysis for CBD detection and validation was based on previously reported methods with modification [12, 19].
- Page 3, Lane 113
- The CBD concentration of each sample is shown in Figure 1.
- Page 3, Lane 115-117
- Of the 7 hemp oil samples tested, 6 contained CBD concentrations ranging from 2.0 ± 0.1 (C6) to 20.5 ± 0.5 (C4) µg/mL, and CBD was not detected in one hemp seed oil sample (C7) or rape oil sample (C9).
- Page 4
- Figure 1. CBD concentrations in CBD oil (a) and hemp seed oil (b) samples.
- Page 4, Page 121-122
- To assess the biological activities of CBD and hemp seed oils in vitro, ABTS and DPPH assays were performed (Figure 1).
- Page 4, Page 132
- Conclusion Discussion
- Page 4, Lane 134-137
- Therefore, unintended intake of CBD from hemp seed oil products might occur. Considering adverse effects of CBD including developmental toxicity, embryo-fatal mortality and neurotoxicity, unintended consumption of CBD should be avoided [22].
- Page 4, Lane 145- Page 5, Lane 150
- Previously, antioxidant activities of cannabinoids and C. sativa extracts were evaluated by DPPH and ABTS assays [18]. CBD and THC showed similar antioxidant activities by ABTS assay, whereas the antioxidant activity of CBD by DPPH assay was two-fold higher than that of THC [18]. Therefore, the ratio of antioxidant activity by DPPH assay to that by ABTS assay might be useful information for quality control of CBD oils to avoid contamination of THC.
- Page 5, Page 152
- Conclusion
- Page 5, Page 153-157
- In this study, we analyzed CBD contents and antioxidant activities of CBD and hemp seed oils commercially available in Japan. The CBD contents of hemp seed oils varied by sample. Because CBD oil samples exhibited strong antioxidant activities, antioxidant assays represent a useful marker for distinguishing between CBD and hemp seed oils. Our results could be applicable to quality control of CBD and hemp seed oil products.
- Page 6, Page 180-182
- [8] Mannucci, C., Navarra, M., Calapai, F., Spagnolo, E., V, Busardò, F., P, Cas R., D., Ippolito, F., M., Calapai G. Neurological Aspects of Medical Use of Cannabidiol. CNS Neurol Disord Drug Targets. 2017, 16 (5), 541-553.
- Page 6, Page 194-195
- [14] Russo, E., B., Marcu, J. Cannabis Pharmacology: The Usual Suspects and a Few Promising Leads. Adv Pharmacol. 2017, 80, 67-134.
- Page 6, Page 206-208
- [18] Hacke, A., C., M., Lima D., de Costa F., Deshmukh K., Li, N., Chow, A., M., Marques, J., A., Pereira, R., P., Kerman, K. Probing the antioxidant activity of Δ9-tetrahydrocannabinol and cannabidiol in Cannabis sativa extracts. Analyst. 2019, 144 (16), 4952-4961.
- Page 6, Page 209-210
- [19] Mandrioli, M., Tura, M., Scotti, S., Toschi, G., T. Fast Detection of 10 Cannabinoids by RP-HPLC-UV Method in Cannabis sativa L. Molecules. 2019, 24 (11), 2113.
- Page 6, Page 217-218
- [22] Huestis, M., A., Solimini, R., Pichini, S., Pacifici, R., Carlier, J., Busardò, F., P. Cannabidiol Adverse Effects and Toxicity. Curr Neuropharmacol. 2019, 17 (10), 974-989.
Reviewer 2 Report
The manuscript of Kitamura et al is potentially interesting but there are many remarks that the authors must take into consideration:
- the authors should provide a broader overview about Cannabidiol, taking also into consideration its adeverse effetcs and toxicity and its current application. The following studies should be included and discussed: Cannabidiol Adverse Effects and Toxicity. Curr Neuropharmacol. 2019;17(10):974-989
- Mannucci et al. Neurological Aspects of Medical Use of Cannabidiol. CNS Neurol Disord Drug Targets. 2017;16(5):541-553.
- the authors should provide more details about method development and validation. About validation, which guidelines were followed ?
- There are several cannabis products containing CBD at different concentrations, and more recently it has been introduced in some countries the so called “Cannabis light”, a cannabis with a very low THC content (< 0.2%) and high CBD percentages. Please discuss this point. I suggest to read the following paper: Determination of cannabinoids in oral fluid and urine of "light cannabis" consumers: a pilot study. Clin Chem Lab Med. 2018 Dec 19;57(2):238-243.
In conclusion, the paper should be improved with an international overview by taking into account all the remarks above reported.
Author Response
I would like to ask you to consider our revised manuscript entitled "Cannabidiol content and in vitro biological activities of commercial cannabidiol oils and hemp seed oils"
According to the reviewers’ comments, we revised the contents as the accompanying sheet. I would appreciate your correspondence.
【2】Comments and Suggestions for Authors
The manuscript of Kitamura et al is potentially interesting but there are many remarks that the authors must take into consideration:
Question 1
the authors should provide a broader overview about Cannabidiol, taking also into consideration its adverse effects and toxicity and its current application. The following studies should be included and discussed:
[21] Huestis MA, Solimini R, Pichini S, Pacifici R, Carlier J, Busardò FP. Cannabidiol Adverse Effects and Toxicity. Curr Neuropharmacol. 2019;17(10):974-989
[8] Mannucci et al. Neurological Aspects of Medical Use of Cannabidiol. CNS Neurol Disord Drug Targets. 2017;16(5):541-553.
Ans.
According to your comment, we cited the references above and discussed adverse effects and toxicity of CBD.
Page 4, Lane 134.
Therefore, unintended intake of CBD from hemp seed oil products might occur. Considering adverse effects of CBD including developmental toxicity, embryo-fatal mortality and neurotoxicity, unintended consumption of CBD should be avoided [22].
Page 6, Lane 181-183, Lane 216-217.
[8] Mannucci, C., Navarra, M., Calapai, F., Spagnolo, E., V, Busardò, F., P, Cas R., D., Ippolito, F., M., Calapai G. Neurological Aspects of Medical Use of Cannabidiol. CNS Neurol Disord Drug Targets. 2017, 16 (5), 541-553.
[22] Huestis, M., A., Solimini, R., Pichini, S., Pacifici, R., Carlier, J., Busardò, F., P. Cannabidiol Adverse Effects and Toxicity. Curr Neuropharmacol. 2019, 17 (10), 974-989.
Question 2
The authors should provide more details about method development and validation. About validation, which guidelines were followed?
Ans.
The HPLC method for CBD detection and validation was based on previously reported methods. According to your suggestion, we described in materials and methods.
Page 2, Lane 64-65
The HPLC analysis for CBD detection and validation was based on previously reported methods with modification [12, 19].
Page 6, Lane 189-191
[12] Citti, C.; Pacchetti, B.; Vandelli, M. A.; Forni, F.; Cannazza, G. Analysis of Cannabinoids in Commercial Hemp Seed Oil and Decarboxylation Kinetics Studies of Cannabidiolic Acid (CBDA). J. Pharm. Biomed. Anal. 2018, 149, 532–540.
Page 6, Lane 209-210
[19] Mandrioli, M., Tura, M., Scotti, S., Toschi, G., T. Fast Detection of 10 Cannabinoids by RP-HPLC-UV Method in Cannabis sativa L. Molecules. 2019, 24 (11), 2113.
Question 3
There are several cannabis products containing CBD at different concentrations, and more recently it has been introduced in some countries the so called “Cannabis light”, a cannabis with a very low THC content (< 0.2%) and high CBD percentages. Please discuss this point. I suggest to read the following paper:
Determination of cannabinoids in oral fluid and urine of "light cannabis" consumers: a pilot study. Pacifici R, et al. Clin Chem Lab Med. 2018 Dec 19;57(2):238-243.
In conclusion, the paper should be improved with an international overview by taking into account all the remarks above reported.
Ans.
We appreciate your suggestion. Unfortunately, we could not obtain the full text and other papers related to “light cannabis”..
[Others]
According to the reviewers’ comments, we clarified our purpose, results and discussion.
All modifications are listed below.
- Page 1, Lane 16-17
- Some products are marketed without a clear distinction between CBD and hemp seed oils.
- Page 1, Lane 25-26
- The results indicate that (1) CBD content varied by hemp seed oil sample and that (2) antioxidant activity could be a useful landmark for discriminating CBD oils from hemp seed oils.
- Page 2, Lane 44-46
- Few reports have been published on CBD quantification of CBD oil products marketed in Japan or on the existence of CBD in hemp seed oils commercially available in Japan. In addition, some products are marketed without a clear distinction between CBD and hemp seed oils.
- Page 2, Lane 50-53
- CBD has demonstrated strong antioxidant effects and can be considered a potent neuroprotective agent [8, 18]. Therefore, in vitro biological assay might be useful for discriminating CBD oils from hemp seed oils. Herein, commercially available CBD and hemp seed oil products were analyzed using high-performance liquid chromatography (HPLC). Additionally, their antioxidant activities were determined.
- Page 2, Lane 57
- Table 1 *Deletion of CBD contents → Figure 1
- Page 2, Lane 64-65
- The HPLC analysis for CBD detection and validation was based on previously reported methods with modification [12, 19].
- Page 3, Lane 113
- The CBD concentration of each sample is shown in Figure 1.
- Page 3, Lane 115-117
- Of the 7 hemp oil samples tested, 6 contained CBD concentrations ranging from 2.0 ± 0.1 (C6) to 20.5 ± 0.5 (C4) µg/mL, and CBD was not detected in one hemp seed oil sample (C7) or rape oil sample (C9).
- Page 4
- Figure 1. CBD concentrations in CBD oil (a) and hemp seed oil (b) samples.
- Page 4, Page 121-122
- To assess the biological activities of CBD and hemp seed oils in vitro, ABTS and DPPH assays were performed (Figure 1).
- Page 4, Page 132
- Conclusion Discussion
- Page 4, Lane 134-137
- Therefore, unintended intake of CBD from hemp seed oil products might occur. Considering adverse effects of CBD including developmental toxicity, embryo-fatal mortality and neurotoxicity, unintended consumption of CBD should be avoided [22].
- Page 4, Lane 145- Page 5, Lane 150
- Previously, antioxidant activities of cannabinoids and C. sativa extracts were evaluated by DPPH and ABTS assays [18]. CBD and THC showed similar antioxidant activities by ABTS assay, whereas the antioxidant activity of CBD by DPPH assay was two-fold higher than that of THC [18]. Therefore, the ratio of antioxidant activity by DPPH assay to that by ABTS assay might be useful information for quality control of CBD oils to avoid contamination of THC.
- Page 5, Page 152
- Conclusion
- Page 5, Page 153-157
- In this study, we analyzed CBD contents and antioxidant activities of CBD and hemp seed oils commercially available in Japan. The CBD contents of hemp seed oils varied by sample. Because CBD oil samples exhibited strong antioxidant activities, antioxidant assays represent a useful marker for distinguishing between CBD and hemp seed oils. Our results could be applicable to quality control of CBD and hemp seed oil products.
- Page 6, Page 180-182
- [8] Mannucci, C., Navarra, M., Calapai, F., Spagnolo, E., V, Busardò, F., P, Cas R., D., Ippolito, F., M., Calapai G. Neurological Aspects of Medical Use of Cannabidiol. CNS Neurol Disord Drug Targets. 2017, 16 (5), 541-553.
- Page 6, Page 194-195
- [14] Russo, E., B., Marcu, J. Cannabis Pharmacology: The Usual Suspects and a Few Promising Leads. Adv Pharmacol. 2017, 80, 67-134.
- Page 6, Page 206-208
- [18] Hacke, A., C., M., Lima D., de Costa F., Deshmukh K., Li, N., Chow, A., M., Marques, J., A., Pereira, R., P., Kerman, K. Probing the antioxidant activity of Δ9-tetrahydrocannabinol and cannabidiol in Cannabis sativa extracts. Analyst. 2019, 144 (16), 4952-4961.
- Page 6, Page 209-210
- [19] Mandrioli, M., Tura, M., Scotti, S., Toschi, G., T. Fast Detection of 10 Cannabinoids by RP-HPLC-UV Method in Cannabis sativa L. Molecules. 2019, 24 (11), 2113.
- Page 6, Page 217-218
- [22] Huestis, M., A., Solimini, R., Pichini, S., Pacifici, R., Carlier, J., Busardò, F., P. Cannabidiol Adverse Effects and Toxicity. Curr Neuropharmacol. 2019, 17 (10), 974-989.
Reviewer 3 Report
Kitamura et al. present an interesting report examining the in vitro activities of several cannabidiol and hemp seed oils. The authors utilize in vitro methods including high-performance liquid chromatography, radical scavenging assays, and enzymatic activity assays. The authors quantify CBD levels within both CBD oils and hemp seed oils, noting a high degree of variability within the hemp products. Additionally, the authors observe dose-dependent antioxidant activity, attributed to the CBD content of the oils. The authors conclude that these methods may be useful for quality control of CBD and hemp seed oils.
In general, the paper is well written. The methods are well described and for the most part, appropriate. While largely observational in nature, the study does provide some insight into the variation in levels of CBD found in various hemp products. The assessment of the biological activity of CBD itself is not novel, as is known to have dose-dependent antioxidant activity. The results of this paper seem to suggest that these methods could be used for quality control, but the novel finding here appears to be the level of variability observed rather than any unexpected differences in biological activities. The critical flaw in this paper is the complete lack of discussion of the results. At no point in the discussion section do the authors cite any other work to validate or explain their results. The authors conclude that the methods could help distinguish between CBD and hemp oils, but there is no discussion of why this would be useful.
Specific comments:
- The introductory paragraph mentions that hemp seeds produce no cannabinoids in their metabolic pathways. It would be helpful to give a reference for this.
- The purpose of the paper should be well described in the introduction. It is not clear that the authors intend these methods to serve as quality control for CBD products.
- No references are provided to justify the use of the enzyme inhibition assays, nor to explain the result. These assays seem to have no value in assessing CBD or hemp oil.
- The use of both the ABTS and DPPH assays seems redundant, as both assays measure antioxidant capacity. No explanation is given to justify the use of both methods. As a dose-dependent response is observed with CBD, it would also be valuable to discuss any similar trends seen in the hemp seed samples.
- Graphs should be provided within the supplementary materials
Author Response
I would like to ask you to consider our revised manuscript entitled "Cannabidiol content and in vitro biological activities of commercial cannabidiol oils and hemp seed oils"
According to the reviewers’ comments, we revised the contents as the accompanying sheet. I would appreciate your correspondence.
Specific comments:
>1. The introductory paragraph mentions that hemp seeds produce no cannabinoids in their metabolic pathways. It would be helpful to give a reference for this.
According to your suggestion, we cited references.
Page 1, Lane 41-43
Considering that hemp seed produces no cannabinoids in its metabolic pathways, it could be contaminated with resin from the flowers and leaves, which contain cannabinoids, during hemp seed processing [12-14].
Page 6, Lane 194-195
[14] Russo, E., B., Marcu, J. Cannabis Pharmacology: The Usual Suspects and a Few Promising Leads. Adv Pharmacol. 2017, 80, 67-134.
>2. The purpose of the paper should be well described in the introduction. It is not clear that the authors intend these methods to serve as quality control for CBD products.
According to your suggestion, we modified “Abstract” and “Indroduction” in the manuscript.
Page 1, Lane 16-17
Some products are marketed without a clear distinction between CBD and hemp seed oils.
Page 1, Lane 24-26
The results indicate that (1) CBD content varied by hemp seed oil sample and that (2) antioxidant activity could be a useful landmark for discriminating CBD oils from hemp seed oils.
Page 2, Lane 44-46
Few reports have been published on CBD quantification of CBD oil products marketed in Japan or on the existence of CBD in hemp seed oils commercially available in Japan. In addition, some products are marketed without a clear distinction between CBD and hemp seed oils.
Page 2, Lane 49-53
CBD has demonstrated strong antioxidant effects and can be considered a potent neuroprotective agent [8, 18]. Therefore, in vitro biological assay might be useful for discriminating CBD oils from hemp seed oils. Herein, commercially available CBD and hemp seed oil products were analyzed using high-performance liquid chromatography (HPLC). Additionally, their antioxidant activities were determined.
>3. No references are provided to justify the use of the enzyme inhibition assays, nor to explain the result. These assays seem to have no value in assessing CBD or hemp oil.
According to your suggestion, we deleted all descriptions of tyrosinase and AchE inhibitory assays.
>4.The use of both the ABTS and DPPH assays seems redundant, as both assays measure antioxidant capacity. No explanation is given to justify the use of both methods. As a dose-dependent response is observed with CBD, it would also be valuable to discuss any similar trends seen in the hemp seed samples.
As you pointed out, both DPPH and ABTS assays measure antioxidant capacity. We described why two anti-oxidant assays were performed as below.
Page 4, Lane 145- Page 5, Lane 150
Previously, antioxidant activities of cannabinoids and C. sativa extracts were evaluated by DPPH and ABTS assays [18]. CBD and THC showed similar antioxidant activities by ABTS assay, whereas the antioxidant activity of CBD by DPPH assay was two-fold higher than that of THC [18]. Therefore, the ratio of antioxidant activity by DPPH assay to that by ABTS assay might be useful information for quality control of CBD oils to avoid contamination of THC.
Page 6, Lane 206-208
[18] Hacke, A., C., M., Lima D., de Costa F., Deshmukh K., Li, N., Chow, A., M., Marques, J., A., Pereira, R., P., Kerman, K. Probing the antioxidant activity of Δ9-tetrahydrocannabinol and cannabidiol in Cannabis sativa extracts. Analyst. 2019, 144 (16), 4952-4961.
>5.Graphs should be provided within the supplementary materials
According to your comment, we showed CBD concentrations in Figure1 in paragraph 3.1.
[Others]
According to the reviewers’ comments, we clarified our purpose, results and discussion.
All modifications are listed below.
- Page 1, Lane 16-17
- Some products are marketed without a clear distinction between CBD and hemp seed oils.
- Page 1, Lane 25-26
- The results indicate that (1) CBD content varied by hemp seed oil sample and that (2) antioxidant activity could be a useful landmark for discriminating CBD oils from hemp seed oils.
- Page 2, Lane 44-46
- Few reports have been published on CBD quantification of CBD oil products marketed in Japan or on the existence of CBD in hemp seed oils commercially available in Japan. In addition, some products are marketed without a clear distinction between CBD and hemp seed oils.
- Page 2, Lane 50-53
- CBD has demonstrated strong antioxidant effects and can be considered a potent neuroprotective agent [8, 18]. Therefore, in vitro biological assay might be useful for discriminating CBD oils from hemp seed oils. Herein, commercially available CBD and hemp seed oil products were analyzed using high-performance liquid chromatography (HPLC). Additionally, their antioxidant activities were determined.
- Page 2, Lane 57
- Table 1 *Deletion of CBD contents → Figure 1
- Page 2, Lane 64-65
- The HPLC analysis for CBD detection and validation was based on previously reported methods with modification [12, 19].
- Page 3, Lane 113
- The CBD concentration of each sample is shown in Figure 1.
- Page 3, Lane 115-117
- Of the 7 hemp oil samples tested, 6 contained CBD concentrations ranging from 2.0 ± 0.1 (C6) to 20.5 ± 0.5 (C4) µg/mL, and CBD was not detected in one hemp seed oil sample (C7) or rape oil sample (C9).
- Page 4
- Figure 1. CBD concentrations in CBD oil (a) and hemp seed oil (b) samples.
- Page 4, Page 121-122
- To assess the biological activities of CBD and hemp seed oils in vitro, ABTS and DPPH assays were performed (Figure 1).
- Page 4, Page 132
- Conclusion Discussion
- Page 4, Lane 134-137
- Therefore, unintended intake of CBD from hemp seed oil products might occur. Considering adverse effects of CBD including developmental toxicity, embryo-fatal mortality and neurotoxicity, unintended consumption of CBD should be avoided [22].
- Page 4, Lane 145- Page 5, Lane 150
- Previously, antioxidant activities of cannabinoids and C. sativa extracts were evaluated by DPPH and ABTS assays [18]. CBD and THC showed similar antioxidant activities by ABTS assay, whereas the antioxidant activity of CBD by DPPH assay was two-fold higher than that of THC [18]. Therefore, the ratio of antioxidant activity by DPPH assay to that by ABTS assay might be useful information for quality control of CBD oils to avoid contamination of THC.
- Page 5, Page 152
- Conclusion
- Page 5, Page 153-157
- In this study, we analyzed CBD contents and antioxidant activities of CBD and hemp seed oils commercially available in Japan. The CBD contents of hemp seed oils varied by sample. Because CBD oil samples exhibited strong antioxidant activities, antioxidant assays represent a useful marker for distinguishing between CBD and hemp seed oils. Our results could be applicable to quality control of CBD and hemp seed oil products.
- Page 6, Page 180-182
- [8] Mannucci, C., Navarra, M., Calapai, F., Spagnolo, E., V, Busardò, F., P, Cas R., D., Ippolito, F., M., Calapai G. Neurological Aspects of Medical Use of Cannabidiol. CNS Neurol Disord Drug Targets. 2017, 16 (5), 541-553.
- Page 6, Page 194-195
- [14] Russo, E., B., Marcu, J. Cannabis Pharmacology: The Usual Suspects and a Few Promising Leads. Adv Pharmacol. 2017, 80, 67-134.
- Page 6, Page 206-208
- [18] Hacke, A., C., M., Lima D., de Costa F., Deshmukh K., Li, N., Chow, A., M., Marques, J., A., Pereira, R., P., Kerman, K. Probing the antioxidant activity of Δ9-tetrahydrocannabinol and cannabidiol in Cannabis sativa extracts. Analyst. 2019, 144 (16), 4952-4961.
- Page 6, Page 209-210
- [19] Mandrioli, M., Tura, M., Scotti, S., Toschi, G., T. Fast Detection of 10 Cannabinoids by RP-HPLC-UV Method in Cannabis sativa L. Molecules. 2019, 24 (11), 2113.
- Page 6, Page 217-218
- [22] Huestis, M., A., Solimini, R., Pichini, S., Pacifici, R., Carlier, J., Busardò, F., P. Cannabidiol Adverse Effects and Toxicity. Curr Neuropharmacol. 2019, 17 (10), 974-989.